# Expression Profiling Reveals the Possible Involvement of the Ubiquitin–Proteasome Pathway in Abiotic Stress Regulation in *Gracilariopsis lemaneiformis*

**DOI:** 10.3390/ijms241512313

**Published:** 2023-08-01

**Authors:** Feng Qin, Guangqiang Shui, Zheng Li, Mengge Tu, Xiaonan Zang

**Affiliations:** Key Laboratory of Marine Genetics and Breeding, Ministry of Education, Ocean University of China, Qingdao 266003, China; qinfeng19971012@gmail.com (F.Q.); gqshui13@163.com (G.S.); lz6780@stu.ouc.edu.cn (Z.L.); 15345360209@163.com (M.T.)

**Keywords:** *Gracilariopsis lemaneiformis*, ubiquitin–proteasome pathway, abiotic stress, gene expression, molecular mechanism

## Abstract

*Gracilariopsis lemaneiformis* is an economically important red macroalga, the cultivation of which is affected by abiotic stresses. This research intends to study the response mechanism of various components of the ubiquitin-protease pathway to abiotic stress in *G. lemaneiformis*. The algae were treated with five common external stresses (high temperature, low temperature, O_3_, PEG, and water shortage) to study the macroscopic and microscopic manifestations of the ubiquitin–proteasome pathway. Firstly, the changes in soluble protein and ubiquitin were detected during the five treatments, and the results showed that the content of soluble protein and ubiquitin significantly increased under most stresses. The content of the soluble protein increased the most on the second day after 20% PEG treatment, which was 1.38 times higher than that of the control group, and the content of ubiquitin increased the most 30 min after water shortage treatment, which was 3.6 times higher than that of the control group. Then, 12 key genes (*E1*, *E2*, *UPL1*, *HRD1*, *UFD1*, *Cul3*, *Cul4*, *DDB2*, *PIAS1*, *FZR1*, *APC8*, and *COP1*) of the ubiquitin–proteasome pathway were studied, including an estimation of the probably regulatory elements in putative promoter regions and an analysis of transcript levels. The results showed that CAAT box, LTR, GC motif, and MBS elements were present in the putative promoter regions, which might have endowed the genes with the ability to respond to stress. The transcript analysis showed that under high temperature, low temperature, PEG, O_3_, and water shortage, all of the genes exhibited instant and significant up-regulation, and different genes had different response levels to different stresses. Many of them also showed the synergistic effect of transcript up-regulation under various stress treatments. In particular, *E1*, *E2*, *Cul3*, *Cul4*, *UPL1*, *HRD1*, and *COP1* performed most significantly under the five stresses. Collectively, our exploration of the ubiquitin–proteasome pathway and the transcript levels of key genes suggest a significant role to cope with adversity, and potential candidate genes can be selected for transformation to obtain stress-resistant strains.

## 1. Introduction

*Gracilariopsis lemaneiformis* is one of the major seaweeds in China, which has important economic value due to its high agar content, many beneficial active substances such as phycobiliproteins and sulfated polysaccharides, and its ability of bioremediation. Wild *G. lemaneiformis* is mainly distributed in the intertidal zone of the Shandong Peninsula, the growth and development of which is significantly affected by abiotic stresses, such as temperature, O_3_, and water shortage, which can cause organelle dysfunction, alterations in cell structure, or cell death. During their long-term evolution, algae have developed various behavioral and molecular strategies to avoid or eliminate the damage caused by abiotic stresses. Studies on algae under various stresses have shown that the antioxidant system, the signal transduction system, the heat shock protein response system, the ubiquitin–protease degradation system, and other functional factors all play an important role in the process of different stresses [1,2,3,4]. Algae often activate the antioxidant systems in response to stress, including antioxidant enzymes that comprise superoxide dismutase, glutathione reductase, catalases, ascorbate peroxidase, and non-enzymatic components that comprise ascorbate, flavonoids, carotenoids, glutathione, tocopherols, and phenols, which can respond to alleviate the damage of free radicals in the early stages of stress [5,6,7]. The presence of the ubiquitin–proteasome pathway is also considered to be an important strategy used to cope with these abiotic stresses. Protein ubiquitination (ubiquitin–proteasome system, UPS) is an important process that is essential for maintaining a balanced level of proteins by degradation of the damaged or surplus proteins. The essential components of protein ubiquitination are ubiquitin, ubiquitin-activating enzyme E1, ubiquitin-binding enzyme E2, ubiquitin-protein ligase E3, and 26S proteasome. More than 80% of intracellular proteins are degraded through their cascade reactions. The ubiquitin–proteasome system participates in all aspects of plant life, including growth and development, circadian rhythm, signal transduction, and abiotic stress [8,9,10,11]. Many reports have shown that the components of the ubiquitin–proteasome system result in enhanced tolerance to abiotic stresses in algae. For example, Ligr found that the ratio of ubiquitin-binding proteins and the rate of ubiquitin synthesis in *Chlamydomonas reinhardtii* increased during cold stress [12]. Tang et al. conducted a transcriptome analysis on pathogen infections in *Porphyra yezoensis* and found that the expression of genes related to the ubiquitin–proteasome system (UPS) significantly increased during infection [13]. There is a difference in ubiquitin in *Saccharina japonica* under normal stress and heat stress: compared with normal conditions, 152 lysine ubiquitination sites in 106 proteins were significantly up-regulated after heat shock [14]. Niaz et al. found that the ubiquitin–proteasome system in *Alexandrium catenella* was activated and significantly up-regulated under low nitrogen and low phosphorus [15]. Li et al. analyzed the expression of the ubiquitin-activating enzyme (E1) and the ubiquitin-binding enzyme (E2) in *G. lemaneiformis* under high temperature stress, which showed that the ubiquitin–proteasome system could be effectively activated to cope with high temperatures [16]. These findings suggest that protein ubiquitination plays an important role in stress resistance in algae. However, research on the ubiquitin–proteasome system in large red algae is still significantly lacking.

In our previous study, the transcriptome sequencing of *G. lemaneiformis* at low temperatures was performed and it was found that most genes of the ubiquitin–protease degradation pathway were significantly up-regulated during low-temperature stress [17]. The genes were also reported to have an essential role in most stresses [18,19,20,21]. The above findings prompted us to undertake an analysis of these genes and check for the abiotic stress regulatory roles of these genes in *G. lemaneiformis*. Therefore, in this study, first, soluble protein and ubiquitin were detected in *G. lemaneiformis* after exposure to various common stresses (high temperature, low temperature, O_3_, PEG, and water shortage) for different durations (0, 1, 2, 3, 6, and 9 days). This study also predicted the putative promoter regions of these genes to explore potential cis-acting elements. In addition, a transcript analysis under a variety of abiotic stress conditions was carried out. Our findings can provide insight into the physiological mechanisms of resistance to stresses, as well as the identification of candidate UPS-related genes that might have roles in inducing stress tolerance and could be exploited for stress manipulation in *G. lemaneiformis*.

## 2. Results

### 2.1. Identification and Analysis of the Upstream Sequences of UPS-Related Genes

The nucleotide sequence of 1 kb upstream of the starting codon ATG was assessed, and a good amount of stress- and signal-responsive elements were discovered (Figure 1). The upstream region of all genes contained at least one CAAT box and ABRE, distributed at different locations upstream of the genes; 92% of the upstream region of the genes contained TGACG-motif and MYB recognition sites, 83% of the upstream region of the genes contained STRE, 75% of the upstream region of the genes contained G-box, 50% of the upstream region of the genes contained MBS, and 33% of the upstream region of the genes contained CAAT box and LTR. Only the upstream region of *E1* contained GARE and AT-rich elements. These elements are all related to abiotic stress, indicating that UPS-related genes may be involved in various abiotic stresses.

### 2.2. Changes in Soluble Protein and Ubiquitin (Ub) under Different Stresses

#### 2.2.1. Changes in Soluble Protein and Ubiquitin (Ub) under High Temperature Stress

Changes in the content of soluble protein and ubiquitin were observed under high temperature (33 °C) stress (Figure 2). In the HT (high temperature) group, both soluble protein and ubiquitin displayed an increasing and then a decreasing trend as the heat stress duration extended. This pattern suggests that short-term exposure to high temperature led to an elevation in ubiquitin content, potentially enhancing the stress resistance of algae. However, prolonged exposure to high temperature caused severe protein denaturation, leading to a decline in soluble protein content. Despite this, the level of ubiquitin remained higher than that of the control group, suggesting its role in assisting the algae in coping with stress. Notably, the highest soluble protein content was observed after two days of heat stress, exhibiting a significant increase of 10.8% compared to the control group (*p* < 0.05). Similarly, the highest ubiquitin levels were recorded after two days of heat stress, showing a substantial increase of 179% compared to the control group (*p* < 0.05). Overall, high temperature exposure induced an increase in both ubiquitin and soluble protein content.

#### 2.2.2. Changes in Soluble Protein and Ubiquitin under Low Temperature Stress

Under low temperature (5 °C) stress (Figure 3), the content of soluble protein and ubiquitin exhibited distinct changes. In the LT (low temperature) group, the content of soluble protein showed a declining trend as the stress duration increased. Conversely, the content of ubiquitin demonstrated an increasing pattern over time, reaching its highest level after nine days of stress, with a significant increase of 247% compared to the control group (*p* < 0.05). Overall, the low temperature stress resulted in an elevation in ubiquitin content. However, this stress condition did not contribute to an increase in soluble protein content.

#### 2.2.3. Changes in Soluble Protein and Ubiquitin under O_3_ Stress

The content of soluble protein and ubiquitin changed under O_3_ stress (Figure 4). The content of soluble protein displayed fluctuations over the stress period, while the content of ubiquitin exhibited a notable and significant increase on the sixth day after stress, surpassing that of the control group (*p* < 0.05).

#### 2.2.4. Changes in Soluble Protein and Ubiquitin under PEG Stresses

Different concentrations of PEG induced distinct changes in the content of soluble protein and ubiquitin (Figure 5). Under 5% and 20% PEG stresses, the content of soluble protein in the treatment group was consistently higher than that of the control group at most time points (*p* < 0.05) (Figure 5a,e). This increase in soluble protein content was in line with the observed elevation in ubiquitin content. Conversely, under 10% PEG stress, the content of ubiquitin increased initially and then decreased with stress duration, mirroring the fluctuations in soluble protein content compared to the control group (Figure 5c,d). In summary, both excessively high and excessively low PEG concentrations led to an increase in both ubiquitin and soluble protein contents.

#### 2.2.5. Changes in Soluble Protein and Ubiquitin under Water Shortage Stress

Distinct responses in the content of soluble protein and ubiquitin were observed under different periods of water shortage stress (Figure 6). The content of ubiquitin started to increase significantly after 20 to 30 min of drought stress (*p* < 0.05), whereas no significant increase was observed during the initial 5 to 10 min of stress. Similarly, the content of soluble protein did not increase until 10 min of drought stress (*p* < 0.05). Overall, the occurrence of water shortage stress triggered an augmentation in both ubiquitin and soluble protein contents.

### 2.3. Transcriptional Regulation of Genes in Response to Different Stress

A heat map depicting gene transcription under each experimental condition was generated (Figure 7), and the expression levels were also represented as bar charts in Appendix A. The gene transcription levels were calculated at different time points under stress conditions relative to the untreated control group at corresponding time points.

In response to high temperature stress, the transcription level of *E2* was immediately activated and continued to increase until 6 h, while *UPL1* started its activation early and continued to increase until 72 h. The transcription level of *E1* was 2.8 and 3.9 times higher than that of the control group at 6 and 24 h, respectively. Similarly, the transcription level of *HRD1* was 3.9 and 3.3 times higher than that of the control group at 6 and 12 h, respectively. Additionally, the transcription level of *COP1* was 9.1 times higher than that of the control group at 24 h.

Under low temperature stress, only *UPL1* displayed an immediate response (1 h), with the transcription level of 8.3 times higher than that of the control group, which was significantly different from the control group. Nearly all genes were up-regulated at 24 and 48 h, and *HRD1* consistently exhibited high expression levels at 6, 12, 24, 48, and 72 h.

Under the treatment of O_3_, the immediate activation of *E1*, *E2*, *UPL1*, *UFD1*, *Cul3*, *Cul4*, and *COP1* in response to O_3_ stress suggests the rapid damage caused by O_3_ to the algae. Subsequently, all genes showed up-regulation at 12 h. Notably, *UPL1*, *Cul3*, *Cul4*, and *COP1* maintained high expression levels during the later stages of stress (48 and 72 h).

In the case of water shortage stress, most genes exhibited immediate up-regulation, but they did not sustain further enhancement, except for *UFD1*. This suggests that severe water shortage stress may inhibit the transcription levels of genes, hindering their continued up-regulation.

To investigate the impact of different stress levels, 5%, 10%, and 20% PEG stresses were used. At equivalent stress time points, an increasing trend of gene expression levels from the 5% PEG to the 20% PEG group was observed. *Cul3*, *Cul4*, and *COP1* consistently exhibited up-regulation in both the 5% and 20% PEG groups. Surprisingly, most genes did not up-regulate as expected under 10% PEG stress, except at 12 h. This observation suggests that an appropriate concentration of PEG might ensure the stability of enzyme proteins, thus weakening the expression of related enzyme proteins of the ubiquitin–proteasome system as a pathway for degrading erroneous or damaged proteins.

Additionally, the Venn diagrams depicting the overlaps of gene transcription in response to high temperature, low temperature, O_3_, and different concentrations of PEG at specific time points are presented (Figure 8a–f). The up-regulated genes in response to various abiotic stress treatments are also listed in Appendix A. Specifically, *E2* exhibited up-regulation at 1 h under all five different stress conditions. At 6 h, *E2*, *HRD1*, *Cul3*, and *Cul4* were up-regulated under all five stress conditions. Similarly, *HRD1*, *Cul3*, *FZR1*, *COP1*, and *APC8* showed up-regulation at 12 h under all five different stress treatments. At 24 h, *E1*, *PIAS1*, and *COP1* were up-regulated in response to all five different stresses, while *COP1* was up-regulated at 48 h under all stress conditions. Additionally, *E2* was up-regulated at 72 h during all five different stress treatments. Furthermore, the overlaps of gene transcription in response to water shortage at different time points are represented in Figure 8g. Under water shortage conditions, *HRD1*, *Cul3*, *Cul4*, *FZR1*, and *COP1* were up-regulated at 5 and 10 min, but all genes exhibited down-regulation at 20 and 30 min, except for *UFD1*.

Furthermore, E1 and E2, the pivotal enzymes involved in the ubiquitin–proteasome system, consistently exhibited high transcription levels under all five stress conditions, especially prominent during the O_3_ stress. This observation aligns with their functional roles in accepting and transferring ubiquitin molecules and ensuring the activation of the related genes encoding E3 and UFD1. The transcription levels of nine E3 ubiquitin ligases are presented in Figure 7. Notably, *UPL1* consistently displayed up-regulation at each time point during high temperature and O_3_ stress treatments. Similarly, *Cul3* and *Cul4* were consistently up-regulated in response to O_3_, 5% PEG, and 20% PEG treatments. *HRD1* maintained up-regulation at 6, 12, 24, 48, and 72 h under low temperature and 5% PEG treatment, as well as at each time point during 20% PEG stress. Moreover, *COP1* sustained up-regulation at 12, 24, and 48 h under high temperature, low temperature, O_3_, and 10% PEG treatments, and it also remained up-regulated at each time point during 5% PEG and 20% PEG conditions. Furthermore, *UFD1*, known for mediating the transfer of ubiquitin to the 26S proteasome, remained up-regulated after exposure to O_3_ for 1, 12, 24, and 72 h, as well as at each time point during 20% PEG stress. The consistent up-regulation of these enzymes, which are integral to the ubiquitin–proteasome pathway and encoded by the identified genes, suggests their crucial positive roles in responding to diverse stresses.

## 3. Discussion

Although ubiquitination plays a crucial role in cellular processes, its investigation in algae remains limited. Abiotic stresses, including high temperature, low temperature, drought, and oxidation, can have severe detrimental effects on algae. These stresses impede algae’s photosynthesis, disrupt chloroplast stability, and induce protein damage and denaturation [22,23]. *G. lemaneiformis*, a large agar-producing algae, is particularly susceptible to these abiotic stresses, leading to a significant reduction in its yield. In response to these abiotic stresses, algae initiate the regulation of gene expression to modulate their growth and development. Therefore, the initial step in comprehending algae’s stress response involves identifying genes that potentially play a role in stress regulation. Through the screening of differentially expressed genes in *G. lemaneiformis* under low temperature stress, it was discovered that the ubiquitin–proteasome pathway plays a pivotal role in this regulatory process [17]. Consequently, this study aims to elucidate the fundamental components of the ubiquitin–protease degradation system, with a specific focus on its efficient regulation under stressful conditions.

Under stress conditions, the up-regulation of most genes in the ubiquitin–protease degradation system may be attributed to the presence of specific regulatory elements, such as the CAAT box and TATA box, located in their promoter regions. These boxes serve as binding sites for the transcription factor CTF/NF-1, which modulates transcription initiation frequency and contributes to the induction of gene expression [24,25,26]. Additionally, the presence of LTR in promoter regions of certain genes indicates their responsiveness to low temperature stress. The activation of *E1*, *UFD1*, *UPL1*, and *COP1* under low temperature stress may be associated with the presence of LTR in their upstream regions. Similarly, genes responding to anoxic inducibility harbor GC motifs in their upstream regions. The activation of *FZR1*, *Cul4*, and *COP1* under ozone stress may be linked to the presence of GC motifs in their promoters. Moreover, genes involved in drought resistance demonstrate the presence of MYB binding sites (MBS), where MYB transcription factors potentially regulate their response to drought stress signals and transcriptional mechanisms. *E1*, *HRD1*, *UFD1*, *Cul3*, *FZR1*, and *PIAS1* possess MBS motifs in their upstream promoter regions, indicating their crucial role in responding to drought stress. Additionally, genes containing ABA response elements (ABRE) and MYB elements are known to enhance resistance to various stresses in plants. *E1*, *E2*, *HRD1*, *UFD1*, *UPL1*, *Cul3*, *Cul4*, *DDB2*, *PIAS1*, *FZR1*, *APC8*, and *COP1* all harbor ABA response elements in their promoter regions, suggesting their significance in responding to multiple stresses. Furthermore, genes responding to osmotic stress and heat shock contain stress-responsive elements (STRE) in their promoter regions. The activation of *E2*, *HRD1*, *UFD1*, *Cul3*, *Cul4*, *DDB2*, *FZR1*, *APC8*, and *COP1* under high temperature, PEG-induced, and water shortage stresses may be attributed to the presence of STRE elements in their upstream regions. Additionally, *E2*, *Cul4*, *DDB2*, and *PIAS1* possess the CAAT box in their promoter regions, which is indicative of their involvement in meristem-related processes and suggest their role in the growth and development of algae, apart from participating in abiotic and biotic stress responses.

The ubiquitin-activating enzyme E1 is the initial enzyme in the ubiquitin–protease degradation pathway, responsible for initiating the protein ubiquitination process. Its primary role involves utilizing ATP-derived energy to activate ubiquitin and transfer it to the ubiquitin-conjugating enzyme E2 [27]. The up-regulation of *E1* under various stress conditions establishes a foundation for the functions of E2, E3, and subsequent enzymes in the ubiquitination process. Serving as the second step in ubiquitination, the ubiquitin-conjugating enzyme E2 receives ubiquitin from the upstream E1, forms the E2-Ub complex, and subsequently transfers it to the E3 ubiquitin ligase [28]. The up-regulated expression of *E2* under diverse stress conditions establishes the foundation for the operation of E3 ubiquitin ligase in the process. The E3 ubiquitin ligase represents the third and final enzyme in the ubiquitination process. Notably, E3 not only associates with E2 but also specifically recognizes various target proteins in the cell requiring ubiquitination. Its primary function involves transferring ubiquitin from E2 to the substrate target protein during the ubiquitination process. As a result, E3 plays a critical role in determining substrate specificity throughout the entire ubiquitination process. It has been established that E3 constitutes a large protein family, with members such as UPL1, HRD1, PIAS1, and COP1 being up-regulated under various stress conditions. These enzymes belong to the HECT type E3 and single RING-finger type E3, which are crucial for accepting ubiquitin from E2. Additionally, the Cul3 and Cul4 protein families, serving as core components and scaffold proteins of the E3 ubiquitin ligase complex, play a vital role in protein ubiquitination and subsequent degradation. This study also identified essential component genes, *Cul3*, *Cul4*, and *DDB2*, as being activated under diverse stress conditions. The APC/C family, a member of the RING family’s E3 ubiquitin ligases, plays a key role in the degradation of cell cycle-related proteins by mediating ubiquitination, thereby regulating cell cycle transitions, post-translational protein modifications, and maintaining balance and stability [29]. In this study, the critical component genes *FZR1* and *APC8* are found to be up-regulated under stress conditions. Furthermore, UFD1, a crucial factor in the regulation of protein degradation by the 26S proteasome, displayed significant up-regulation under various stress conditions. Collectively, these findings highlight the collaborative involvement of multiple components within the ubiquitin–protease degradation system under stress conditions, leading to enhanced ubiquitination level. This observation is consistent with the observed increase in ubiquitin content during various stress conditions.

Although the ubiquitin–proteasome pathway in algae remains underexplored, studies in plants and some algae have revealed the up-regulation of certain components involved in the ubiquitination process during stress conditions. For instance, Kaur et al. observed up-regulation of the ubiquitin-activating enzyme E1 during waterlogging stress in maize [30]. Similarly, *Lemna minor* L. exposed to high temperatures showed increased Ub-protein conjugates and transcription levels of E1 [31], suggesting a positive role for E1 in stress responses. The E2 ubiquitin-conjugating enzyme (UBC18) was found to regulate drought and salt stress responses in *Arabidopsis thaliana* by controlling downstream genes of ERF1 [32], and overexpression of mung bean E2 enhanced osmotic stress tolerance in *Arabidopsis thaliana* [33]. In the intertidal green macroalga *Ulva fasciata*, the transcript level of E2 increased during high salinity stress [34]. Consistent with these findings, E2 in *G. lemaneiformis* also exhibited activity under temperature, osmotic pressure, and oxidative stresses, indicating its involvement in stress responses. Ubiquitin-protein ligase E3, as reviewed by Shu and Yang, played essential roles in plant development and abiotic stress responses [35]. Various ubiquitin-protein ligase, including HECT type E3s, single RING-finger type E3s, APC-subtype E3s, and Cullin-box E3s, are effectively activated during drought, osmotic, salinity, and temperature stresses. UPLs, such as UPL1, UPL3, and UPL5, have been shown to play significant roles in plant immunity, with mutations in these genes reducing immune responses activated by salicylic acid [36,37,38]. UPL1 was found to be active during various stress treatments in *G. lemaneiformis*. HRD1 is involved in plant cuticular lipid biosynthesis and provides protection against biotic and abiotic stresses [39]. The disruption of the HRD1/HRD3 complex in *Arabidopsis thaliana* increased plant sensitivity to salt and led to the accumulation of ERAD substrates, implying HRD1’s involvement in protein degradation and salt stress resistance [40,41]. COP1, which regulates light signaling and plant growth and development, was found to contribute to stability in *Arabidopsis thaliana* under low temperature environments and positively contribute to salt stress resistance [42,43,44]. In this study, COP1 also exhibited positive participation in various abiotic stresses. Other E3 ubiquitin ligases have distinct functions. For example, CRLC (Cullin-RING ligase)3-based E3 ligases are involved in ABA biosynthesis, signaling, and downstream responses [45]. CUL4 E3 complexes regulate various processes in plants, such as photomorphogenesis, flowering, and abiotic stress response [46]. Additionally, DDB2 participated in nucleotide excision repair, UV response, and maintenance of genomic integrity [47,48]. The active expression of these E3 ubiquitin ligases plays a crucial role in degrading damaged proteins and maintaining cellular stability. Furthermore, UFD1 in grapevine was shown to be involved in responses to salt stress, light treatment, and pathogen infection during grapevine growth [49]. CDC48 and its cofactor UFD1 were found to be essential for regulating proteins and responding to oxidative stress [50]. In this study, UFD1 was up-regulated under most stress conditions, facilitating the efficient delivery of ubiquitin proteins to the 26S proteasome.

In addition, PEG was used as a drought regulator, which could cause cell dehydration and alter the structures of the plasma membrane to induce drought stress [51]. At different PEG concentrations, the genes’ response was also different. Both 5% and 20% PEG treatments caused up-regulation of genes related to the ubiquitin–proteasome degradation pathway. In comparison, the treatment with 10% PEG did not induce a positive gene response in the later stages. In addition to functioning as an osmotic regulator that could simulate adverse conditions such as drought, PEG could interact with proteins and provide a protective effect on them. Karla et al., found that PEG could protect the enzymatic activity of horseradish peroxidase (HRP), versatile peroxidase (VP), and commercial Coprinus peroxidase (BP) [52]. Jun-Beom et al., found that the attachment of PEG to cocaine esterase (CocE) protected CocE against thermal degradation and protease digestion [53]. Li et al., found that the combination of nanosilicasol and polyethylene glycol (PEG) 1000 could enhance the thermostability of beta-cyclodextrin glycosyltransferase in *Bacillus circulans* [54]. Others also described the potential positive effects of PEG [55,56]. Therefore, we speculated that when PEG reaches a specific concentration, the stability of enzymatic proteins could be improved, and the normal activity of the enzymes is guaranteed. The possibility of protein misfolding is reduced, which may prevent the activation of the ubiquitin–protease degradation system and the expression of genes involved in the ubiquitin–protease degradation system. When the concentration of PEG becomes too low or exceeds a certain amount, it could induce stress in algae. Dong studied the effect of PEG on the growth of onions and it was found that the 10% and 20% PEG treatments had a more pronounced inhibitory effect compared to the 15% PEG treatment [57].

Based on the transcription levels of genes associated with the ubiquitin–protease degradation system, it was observed that these genes were uniformly up-regulated under various stresses, indicating coordinated response and potential synergy (Figure 9). Specifically, E1 and E2 showed up-regulation during all five stress conditions (high temperature, low temperature, O_3_, PEG, and water shortage). Several genes belonging to different E3 families, such as Cullin-box E3 (Cul3 and Cul4), HECT type E3 (UPL1), and single RING-finger E3 (HRD1 and COP1), were up-regulated under multiple stress conditions. Additionally, UFD1 exhibited up-regulation in response to most stress conditions. These findings suggest that the proteins encoded by these genes likely play crucial roles in responding to environmental stresses in *G. lemaneiformis*. The observed synergistic mechanisms among various components of the ubiquitin–proteasome pathway in *G. lemaneiformis* are akin to what has been observed in plants. Furthermore, it was found that the content of ubiquitin increased, and then the content of soluble protein increased under most stress conditions. However, under low temperature and O_3_ stress, the content of soluble protein showed a downward trend, possibly attributed to severe stress leading to increased protein denaturation.

## 4. Materials and Methods

### 4.1. Algal Strains and Culture Conditions

Wild tetrasporophytes of *G. lemaneiformis* were collected from the intertidal zone of Fu shan Bay (36.0° N, 120.3° E), Qingdao, China in October 2022. The algae were thoroughly brushed to remove any epiphytes. They were cultured in tanks with sterile seawater enriched with f/2 medium (the composition of f/2 medium is provided in Appendix A) at 21 °C under a 12 L:12 D photoperiod 20 µmol photons m^−2^ s^−1^ light intensity.

### 4.2. Nucleotide Sequence Retrieval and Putative Promoter Analysis of UPS-Related Genes

Based on the transcriptome database of *G. lemaneiformis* at low temperature [17], 12 differentially expressed genes in the ubiquitin–proteasome pathway (pathway number: ko04120) (Appendix A) were identified: *E1* (ubiquitin-activating enzyme E1), *E2* (ubiquitin-conjugating enzyme E2), *UPL1* (E3 ubiquitin-protein ligase UPL1), *HRD1* (E3 ubiquitin-protein ligase HRD1), *UFD1* (ubiquitin fusion degradation protein 1 UFD1), *Cul3* (Cul3), *Cul4* (Cul4), *DDB2* (DNA damage-binding protein 2), *PIAS1* (E3 SUMO-protein ligase PIAS1), *FZR1* (cell division cycle 20-like protein 1, cofactor of APC complex), *APC8* (anaphase-promoting complex subunit 8), and COP1 (E3 ubiquitin-protein ligase COP1). To explore the possible correlation between the up-regulation of these UPS-related genes during various stresses and the presence of corresponding stress-responsive elements in their putative promoter regions, we performed an analysis of the 1 kb nucleotide sequence upstream of each gene using the Plant Cis-Acting Regulatory Elements (Plant CARE) database [58]. The putative promoter sequences of genes are presented in the Appendix A.

### 4.3. Abiotic Stress Treatments and Tissue Sampling

After a 7-day adaptive culture in the laboratory, the algae were subjected to five different abiotic stress treatments, including heat stress (33 °C), cold stress (5 °C), PEG-8000 stress (5%, 10%, and 20%), O_3_ stress (1–2 g/h), and water shortage stress (5, 10, 20, and 30 min). Except for the above stress factors, all other cultivation conditions remained consistent. The algae branches were collected as three biological replicates at the same time points of 0, 1, 2, 3, 6, and 9 days for the detection of soluble protein and ubiquitin at 0, 1, 6, 12, 24, 48, and 72 h for the analysis of the transcript levels. The untreated control group was also sampled at the same time as the treatment group. The extracts were prepared using 0.3 g samples in 2.7 mL of 0.9% sodium chloride solution. The homogenate was centrifuged at 2500× *g* for 10 min at 4 °C, and the supernatants were used to measure the contents of soluble protein and ubiquitin. Algae (0.1–0.15 g) were sampled and then placed into liquid nitrogen for freezing. They were stored at −80 °C for RNA extraction and analysis of the transcript levels of the 12 UPS-related genes.

### 4.4. The Content of Soluble Protein

The content of soluble protein was determined using a spectrophotometric assay, following the instructions provided in the TP detection Kit manual (Nanjing Jiancheng Bioengineering Institute, Nanjing, China). Soluble protein determination is based on the Bradford method/reagent [59], using the staining method with Coomassie Brilliant Blue G250. In short, the solution will turn blue when the brownish red Coomassie brilliant blue combines with the -NH3^+^ of the protein. Absorption spectrophotometric values at a wavelength of 595 nm were measured and used to calculate the total protein content using the following formula:$$\mathrm{TP}\;(\mathrm{mg}/g\;\mathrm{FW})=\frac{A_{sample}-A_{control}}{A_{standard}-A_{control}}\times C_{standard}\times 2.7/0.3$$
where the A_control_ is the absorbance of the distilled water, the A_sample_ is the absorbance of the extract, and A_standard_ is the absorbance of the standard protein. C_standard_ is the concentration of the standard protein (fetal bovine serum) (0.524 g/L).

### 4.5. Ubiquitin (Ub) Determination

Ubiquitin (Ub) was evaluated using the double antibody sandwich method, following the instruction manual of the Plant Ub ELISA Kit (Shanghai Fusheng Industrial Co., Ltd., Shanghai, China). Briefly, the antibody–antigen enzyme-labeled antibody complex will form when plant-Ub is added to a microwell plate containing a double antibody sandwich. After washing and coloring, the solution will turn yellow. The depth of color and the concentration of Ub in the sample were positively correlated. The absorption spectrophotometric values at a wavelength of 450 nm were measured and used to calculate the concentration of Ub in the sample using the standard curve.

### 4.6. RNA Extraction and cDNA Synthesis

Total RNA from all samples, under native and different conditions, was extracted following the instruction manual of the HP Plant RNA Kit (Omega Bio-Tek, Norcross, GA, USA). The quality of the RNA samples was assessed using agarose gel electrophoresis and NanoDrop, measuring the OD260/280 and OD260/230 ratios. The cDNA was synthesized according to the instruction manual of the Prime Script™ RT Master Mix (Perfect Real Time) (Takara Bio, Kusatsu, Japan). The synthesized cDNA was stored at −40 °C.

### 4.7. The Transcript Analysis of Genes

The qRT-PCR (TB Green Premix Ex TaqTM II Kit, Takara Bio, Kusatsu, Japan) was used to analyze the transcript levels of genes. Reaction steps followed the program [60]: 95 °C for 20 s, followed by 40 cycles of 95 °C for 15 s, 55 °C for 15 s, 72 °C for 20 s, and reading the fluorescence signal, followed by 1 cycle of 95 °C for 15 s, 60 °C for 60 s, and 95 °C for 15 s. Each reaction was performed in three biological replicates and three technical replicates. The RT-PCR was performed using the LightCycler^®^ 96 System from Roche (Mannheim, Germany). The transcription of each gene in the control group at the same time point was used as a calibrator to determine the expression level of genes in the treatment groups. The amount of cDNA for each PCR sample was approximately 300 ng/μL; the β-actin gene (*ACT*) from *G. lemaneiformis* was used as the reference gene. All primers were designed using the Primer 5.0 software (Primer, Toronto, ON, Canada) and are listed in Table 1. The relative transcript level was calculated using the 2^−∆∆Ct^ method [61].

### 4.8. Statistical Analysis

In the data analysis of soluble protein and ubiquitin of *G. lemaneiformis* under different stresses, separate one-way analysis of variance (ANOVA) was used and followed by pairwise comparisons at different groups using the LSD method (*p* = 0.05). All of the data were expressed as the mean ± standard deviation (SD).

## 5. Conclusions

In this study, the ubiquitin–proteasome pathway was found to play a pivotal role in the response of *G. lemaneiformis* to various common stresses, including high temperature, low temperature, O_3_, PEG, and water shortage. The up-regulation of most Ub-related genes, such as *E1*, *E2*, *Cul3* (Cullin-box E3), *Cul4* (Cullin-box E3), *UPL1* (HECT type E3), *HRD1*, *COP1* (single RING-finger type E3), and *UFD1*, was observed, consistently coinciding with the increase in ubiquitin content. These findings provide valuable insights into the functional roles of genes involved in the ubiquitin–proteasome pathway, and the identified potential candidate genes may hold promise for enhancing the stress tolerance of algae to various adverse conditions.

## Figures and Tables

**Figure 1 ijms-24-12313-f001:**
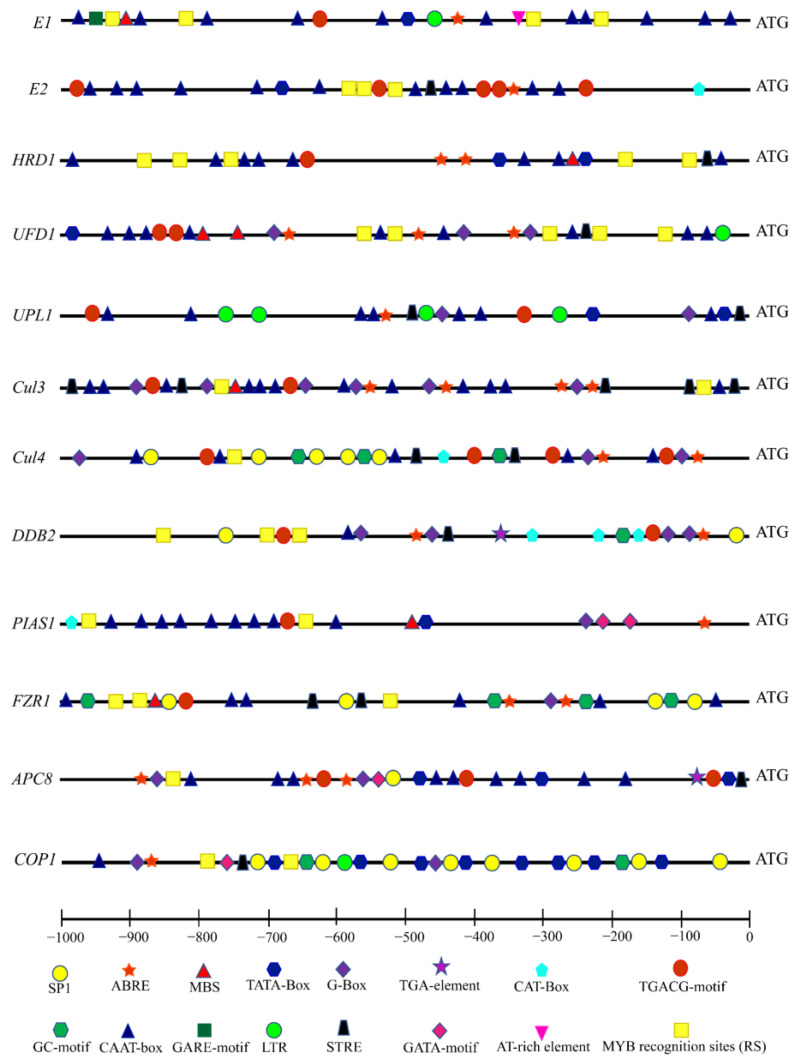
The analysis of cis-acting elements in putative promoter sequences of the UPS genes. The nucleotide sequence ≤1 kb upstream of the ATG of each gene carries multiple stress and signal-responsive elements. A scale at the bottom indicates the approximate location of corresponding elements. Each element is shown with a different shape and color, which is described at the bottom of the figure. Each gene name is displayed on the left.

**Figure 2 ijms-24-12313-f002:**
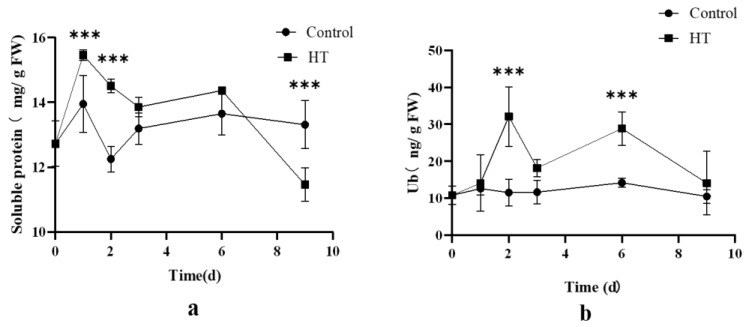
The content of soluble protein and Ub in high temperature (33 °C) treatment group and control group. (**a**) The content of soluble protein. (**b**) The content of Ub. The data are expressed as the mean ± standard deviation (SD), *** indicates a significant difference from the control group (*p* < 0.05).

**Figure 3 ijms-24-12313-f003:**
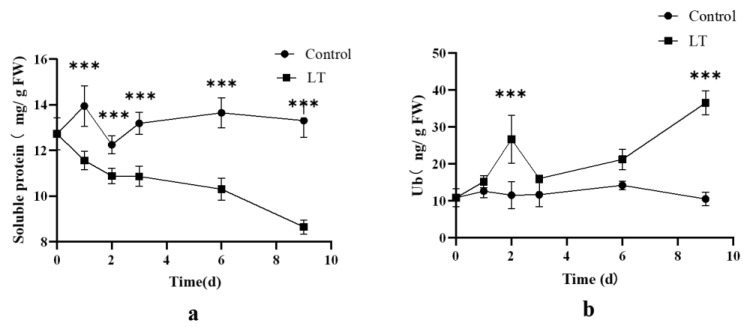
The content of soluble protein and Ub in low temperature (5 °C) treatment group and control group. (**a**) The content of soluble protein. (**b**) The content of Ub. The data are expressed as the mean ± standard deviation (SD), *** indicates a significant difference from the control group (*p* < 0.05).

**Figure 4 ijms-24-12313-f004:**
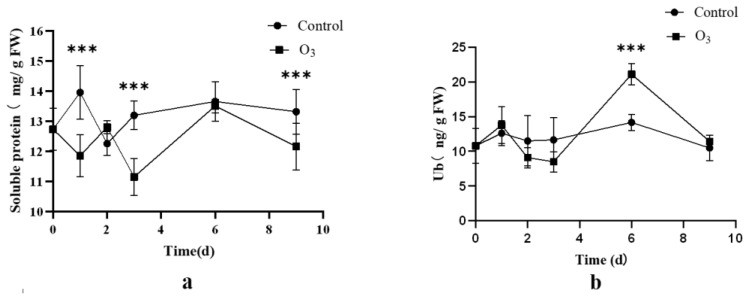
The content of soluble protein and Ub in O_3_ treatment group and control group_._ (**a**) The content of soluble protein. (**b**) The content of Ub_._ The data are expressed as the mean ± standard deviation (SD), *** indicates a significant difference from the control group (*p* < 0.05).

**Figure 5 ijms-24-12313-f005:**
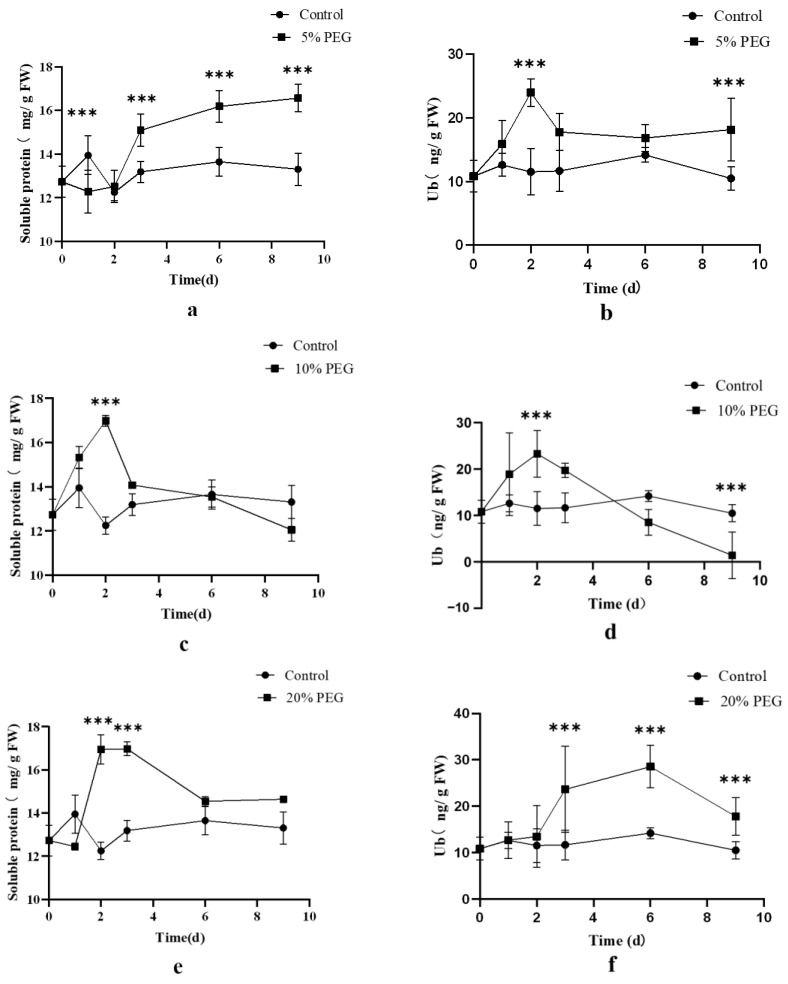
The content of soluble protein and Ub in different concentrations of PEG groups and control group. (**a**) The content of soluble protein in 5% PEG group and control group. (**b**) The content of Ub in 5% PEG group and control group. (**c**) The content of soluble protein in 10% PEG group and control group. (**d**) The content of Ub in 10% PEG group and control group. (**e**) The content of soluble protein in 20% PEG group and control group. (**f**) The content of Ub in 20% PEG group and control group. The data are expressed as the mean ± standard deviation (SD), *** indicates a significant difference from the control group (*p* < 0.05).

**Figure 6 ijms-24-12313-f006:**
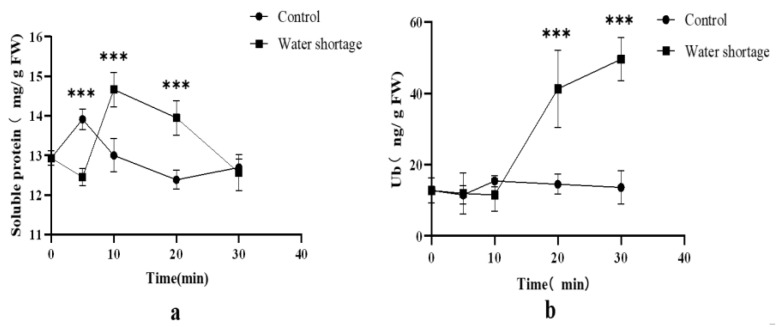
The content of soluble protein and Ub in water shortage group and control group. (**a**) The content of soluble protein. (**b**) The content of Ub. The data are expressed as the mean ± standard deviation (SD), *** indicates a significant difference from the control group (*p* < 0.05).

**Figure 7 ijms-24-12313-f007:**
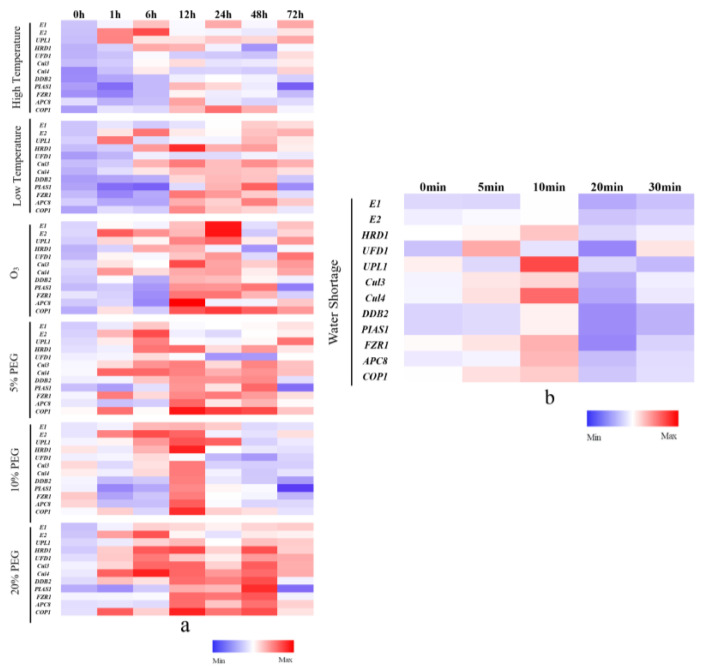
Cluster analysis of UPS genes at different conditions and different times. The vertical axis stands for relative transcription of different genes at different conditions, and the color represents the transcription level of the gene. The horizontal axis stands for different time points. (**a**) The cluster analysis of UPS genes under high temperature, low temperature, O_3_, 5%, 10%, and 20% PEG treatments. (**b**) The cluster analysis of UPS genes under water shortage treatment.

**Figure 8 ijms-24-12313-f008:**
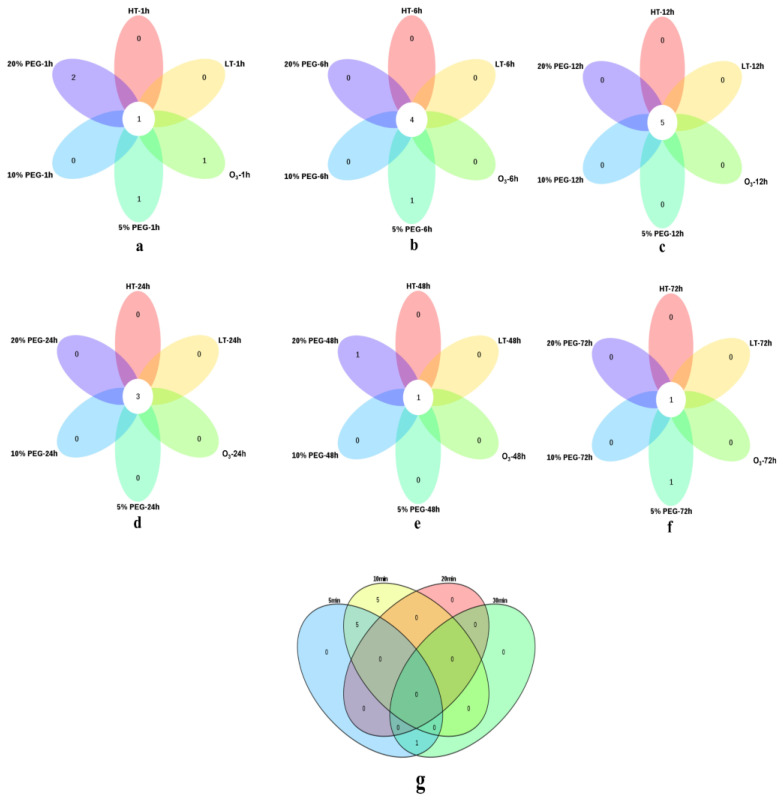
Venn diagram of the up-regulated genes under different conditions. (**a**–**f**) represents the overlaps of up-regulated genes at each time point under high temperature, low temperature, O_3_, 5% PEG, 10% PEG, and 20% PEG treatments. (**g**) represents the overlaps of up-regulated genes at each time point during water shortage.

**Figure 9 ijms-24-12313-f009:**
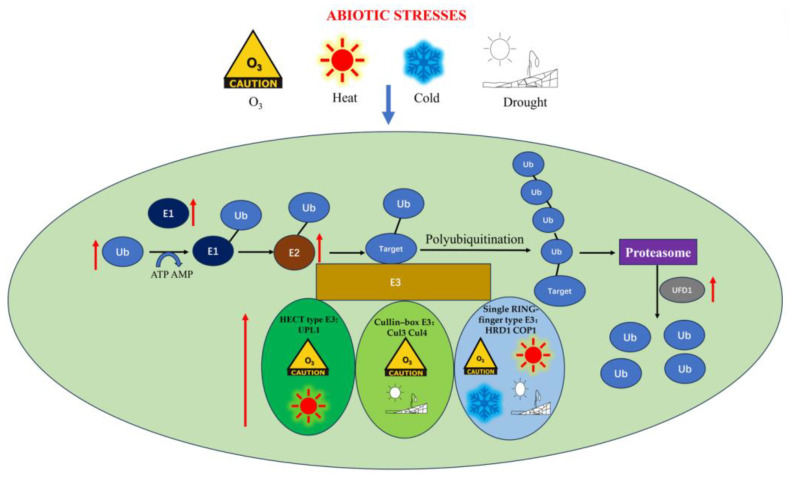
The ubiquitin–proteasome pathway in *G. lemaneiformis*. The red arrow indicates that both E1 and E2, as well as different elements of E3 and UFD1, exhibit positive effects and have synergistic effects under stress conditions. Different signals indicate various stress conditions. Signs and elements of E3 in the same box indicate that these elements may play important roles under these stresses.

**Table 1 ijms-24-12313-t001:** List of qPCR genes and primers.

Gene Name	Primer
*ACT* (actin)	F: CTACTCGTTTACCACTTCTGCTGA
	R: TTCCATTCCGACCAACTCTG
*E1* (ubiquitin-activating enzyme E1)	F: CACAGTCGTTTGCCATCAGC
	R: TCTTCCCCACATCCCACCTA
*E2* (ubiquitin-conjugating enzyme E2)	F: TTATCCGTTCAAGCCACCG
	R: GAGCGATTTCAGGCACAAG
*UPL1* (E3 ubiquitin-protein ligase UPL1)	F: GCGAAGGGGTTTGGAAATAAT
	R: TGTCTGTAGACTTGACAGGAGGGAT
*HRD1* (E3 ubiquitin-protein ligase HRD1)	F: TTCCTCAGGTATCGCCGTGTT
	R: GTTGCGGTTGTGCTTGGTTCT
*UFD1* (ubiquitin fusion degradation protein 1)	F: CACCTTCAGCACTCGACTCCTT
	R: CGGCGATAAACTCCTGTACTCC
*Cul3* (Cul3)	F: GGATGCTGCGAAATGATAAGGT
	R: ATTCTTGGGAGATGGGGATGGT
*Cul4* (Cul4)	F: GGGATATGGGCTTGCTTCTGTT
	R: TGCGGTAGTAGGTGTCGGTTGA
*DDB2* (DNA damage-binding protein 2)	F: CACAATTCCCCTCCTCAACTC
	R: TGAACAGACTGACGCTTCCCT
*PIAS1* (E3 SUMO-protein ligase PIAS1)	F: CCAGTCAAAGGCAAAAGGTGTC
	R: CCTCCATATCATCGTCTTCGTCA
*FZR1* (cell division cycle 20-like protein 1, cofactor of APC complex)	F: CACATTCTGTCCAACGCCACT
	R: GTTCCAAGCAACAGAGCATACA
*APC8* (anaphase-promoting complex subunit 8)	F: AACGACGAAAAGAAGAAGAGCG
	R: CCAGTGGTTGTCAATGTCCAAA
*COP1* (E3 ubiquitin-protein ligase COP1)	F: TCACGGCATTCATTACTACG
	R: GCACCAACAAACGCTACTC

## Data Availability

Not applicable.

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
