# Peer review of "Expression Profiling Reveals the Possible Involvement of the Ubiquitin–Proteasome Pathway in Abiotic Stress Regulation in Gracilariopsis lemaneiformis"

_ijms, 2023, doi:10.3390/ijms241512313_

Round 1

Reviewer 1 Report

The title "Comprehensive expression profiling reveals the possible involvement of ubiquitin-proteasome pathway in stress regulation in Gracilariopsis lemaneiformis" is overly broad and lacks specificity. It would be beneficial to include more specific information about the stressors and the specific aspects of stress regulation that were investigated. The abstract is poorly structured and lacks clarity. It fails to provide a clear overview of the research objectives, methods employed, and key findings. The abstract should include a concise statement of the problem being addressed, the experimental design, the major results, and the significance of the findings. The introduction should provide a more comprehensive literature review. The authors should discuss the current understanding of stress responses in macroalgae and the involvement of the ubiquitin-proteasome pathway. This would help establish the context and significance of the study. The methods section is insufficiently described. Important details regarding the experimental design, stress treatments, sampling techniques, and statistical analyses are missing. Without this critical information, it is challenging to evaluate the reliability and reproducibility of the results. The results presented in the abstract are too general and lack specific quantitative data. The authors should provide more specific information on the observed changes in soluble protein and ubiquitin levels under each stress treatment. Including statistical analyses and specific values would strengthen the findings. The discussion is limited and does not provide a thorough analysis or interpretation of the results. The authors should discuss the implications of the findings in relation to existing knowledge and highlight the novelty and significance of their results. Additionally, they should address any limitations or potential alternative explanations for the observed patterns.

English is fine.

Author Response

Thanks very much for your comments concerning our manuscript entitled “Comprehensive expression profiling reveals the possible involvement of ubiquitin-proteasome pathway in stress regulation in Gracilariopsis lemaneiformis” (Manuscript: ijms-2474555). Those comments are valuable and helpful for improving our paper, as well as have guiding significance to our research. We have studied these comments carefully and have made a thorough revision of our manuscript. Revised portion are marked in red in the manuscript.

       The attachment is our description on revision according to the reviewer’s comments.

Reviewer 2 Report

The manuscript entitled “Comprehensive Expression Profiling Reveals the Possible Involvement of Ubiquitin-Proteasome Pathway in Stress Regulation in Gracilariopsis lemaneiformis” focus on the ubiquitin-proteasome pathway in macroalga. In general, the manuscript is clearly written and well organised but there are two major issues.

Firstly, the novelty and the overall merit of the manuscript are low.  Even in the introduction authors say: “These findings suggest that protein ubiquitination plays an important role in stress resistance in algae.” It is true the involvement of the ubiquitin-proteasome pathway both in plants and in algae is well-known. And the data presented here do not add anything really new to the existing knowledge. Moreover, the diversity of experiments is very low. The authors analysed only soluble protein content and the level of ubiquitin. Moreover, as claimed by authors for example in line 18 “intensive study” of genes related to the ubiquitin-proteasome pathway is in fact limited to promoter analysis and gene expression determination. Gracilariopsis lemaneiformis - the importance of this macroalga is also limited to some selected regions of the world. The manuscript neither provides any novel insight into the ubiquitin-proteasome pathway nor proposes any novel mechanism.    

Secondly and more importantly, the is no proper control in the presented experiments. It is not enough to collect material at 0 point (i.e. at the start of the experiment) and then only from treatments. At the same time points of treatment also materials from control (i.e. alga grown without tested stress factor) should be collected. Otherwise, it is possible the observed changes are due to for example circadian rhythm or another unknown event.   

Specific comments are in the file attached.  

The English in general is OK. Some checking of spelling and style is needed. 

Author Response

      Thanks very much for your comments concerning our manuscript entitled “Comprehensive expression profiling reveals the possible involvement of ubiquitin-proteasome pathway in stress regulation in Gracilariopsis lemaneiformis” (Manuscript: ijms-2474555). Those comments are valuable and helpful for improving our paper, as well as have guiding significance to our research. We have studied these comments carefully and have made a thorough revision of our manuscript. Revised portion are marked in red in the manuscript.

 Here below is our description on revision according to the reviewer’s comments.

Point 1: Firstly, the novelty and the overall merit of the manuscript are low. Even in the introduction authors say: “These findings suggest that protein ubiquitination plays an important role in stress resistance in algae.” It is true the involvement of the ubiquitin-proteasome pathway both in plants and in algae is well-known. And the data presented here do not add anything really new to the existing knowledge. Moreover, the diversity of experiments is very low. The authors analysed only soluble protein content and the level of ubiquitin. Moreover, as claimed by authors for example in line 18 “intensive study” of genes related to the ubiquitin-proteasome pathway is in fact limited to promoter analysis and gene expression determination. Gracilariopsis lemaneiformis - the importance of this macroalga is also limited to some selected regions of the world. The manuscript neither provides any novel insight into the ubiquitin-proteasome pathway nor proposes any novel mechanism.   

Response 1: Thanks for your suggestion. Although the ubiquitin-proteasome pathway has been studied widely and a large amount of data has been obtained in many plants, due to species differences, this pathway has not been well understood in large red algae and it still requires a lot of work to be carried out. As the second most importantlly economic red algae in China, Gracilariopsis lemaneiformis is widely distributed in China's sea areas and has high application value. However, there are very few related studies in Gracilariopsis lemaneiformis, and the related elements of ubiquitin-proteasome pathway have not been reported. This work is the first time to report the related genes of ubiquitin-proteasome pathway from Gracilariopsis lemaneiformis and analyzed their expression levels under common stress conditions, and we found that the ubiquitin pathway really plays an important role in the process of stress resistance in Gracilariopsis lemaneiformis. Chinese researchers have been committed to studying the stress resistance mechanism and breeding work of Gracilariopsis lemaneiformis, hoping to cultivate new varieties of Gracilariopsis lemaneiformis with strong stress resistance. Therefore, the research in this work is crucial for the breeding of Gracilariopsis lemaneiformis in the future.

  Due to the lack of a genetic transformation system for Gracilariopsis lemaneiformis, we are unable to study the function of related gene elements in the ubiquitin-proteasome pathway directly by means of knock-in or knock-out. Therefore, we temporarily use real-time quantitative methods to study gene expression levels. In the future, we will commit to establishing a genetic transformation system for Gracilariopsis lemaneiformis, as well as using heterologous expression method to further study the functions of related genes. Due to the limited research methods currently available, as the reviewer pointed out, it is indeed not yet intensive enough. Therefore, we have removed the word 'intensive' from the manuscript.

Point 2: Secondly and more importantly, the is no proper control in the presented experiments. It is not enough to collect material at 0 point (i.e. at the start of the experiment) and then only from treatments. At the same time points of treatment also materials from control (i.e. alga grown without tested stress factor) should be collected. Otherwise, it is possible the observed changes are due to for example circadian rhythm or another unknown event.  

Response 2: Thanks for your advice, we have collected sample from control group at the same time points. Because the changes of various indicators from the control group are not significant, the previous version of the manuscript did not include data for the control group, now the data for the control group in the manuscript has been added.

The attachment file“Response to reviewer2 about detailed comments in manuscript” is the response to the reviewer’s detailed comments in manuscript.

Reviewer 3 Report

Authors present interesting and import ant study related to the analysis of ubiquitin-proteasome pathway genes participating in stress regulation in Gracilariopsis lemaneiformis. Study is generally good planned and performed. Results are new and significant. Following minor comments sholud be addressed before the publication:

 Typographical errors-check the entire text:

Line 11- Capitalize „ubiquitin”

244- should be degradation

299- do not capitalize „exposure”

331- Arabidopsis thalian- write in italics

334- should be Cullin not ullin

399- put space after stress

Section 4.2

How Authors assured that obtained sequences are true promotor regions? Did Authors use a database containing such DNA sequences? Provide the source of promoter sequences.

Section 4.4

Authors should add that the staining metod used Coomassie brilliant blue G250. Putatively soluble protein determination is based on Bradford metod/reagent. If so, add the citation of original Bradford method:

Bradford, M.M. (1976) A rapid and sensitive method for the quantitation of microgram quantities of protein utilizing the principle of protein-dye binding. Anal. Biochem. 72, 248–254.

Line 418- provide the name of standard protein.

Section 4.6

Was the DNaseI used to remove remnants of genomic DNA from RNA samples.

Section 4.7

Provide amount of RNA/cDNA per one PCR sample

Provide information from which species originates the ACT reference gene.

Manufacturer of the RT-PCR equipment- provide its name and nationality.

Minor editing of English language required.

Author Response

(The authors gave the same response as above.)
